# Exploring STEM Teacher Learning in Out-of-School Settings: Sites for Learning about Relevance

**Ti'Era Worsley *** and **Sara Heredia**

Department of Teacher Education and Higher Education, The University of North Carolina at Greensboro, Greensboro, NC 27412, USA
* Correspondence: tdworsle@uncg.edu

**Abstract:** Decades of research on student's science, technology, engineering, and mathematics (STEM) learning have highlighted the need to make science content more relevant for students. However, science teachers have described this as difficult, with large numbers of students in their classrooms and an administrative focus on achievement in high-stakes assessments. In this study, we explore the learning opportunities for two in-service science teachers when they participated in an out-of-school professional development (PD) program to design and implement a STEM after-school program (ASP) in partnership with a local university. We specifically look at how participation in the PD and the implementation of an ASP supported them in creating science content that was relevant to their students, as well as the activities and tools teachers highlighted as important for their learning. We provide a rich description through case study methodology of how both teachers learned to make the science content more relevant to students as they developed an ASP about water issues within the community. We argue that the ASP became a productive space for teachers to work with students in informal settings and acquaint themselves with their students in new and different ways to those afforded by their formal classroom.

**Keywords:** STEM; teachers; informal learning; relevance

## 1. Introduction

A key assumption of reform documents in science education is that students learn science best when it connects to their lived experiences [1]. While most equity-based pedagogies focus on understanding and leveraging students' assets and resources for their science learning, few descriptions of what this looks like in formal STEM contexts describe how and in what ways science teachers can develop this knowledge of their students. For example, early-career teachers that participated in an induction course designed to develop equity-based instructional practices expressed changes to their ideas about the community and the resources that students bring [2]. This included an exploration into teachers' ideas about students from historically marginalized groups. The teachers expressed changes to their ideas about the community and resources that students bring but were challenged with how to access this information about their students given their class sizes and lack of resources to do this work. We posit that out-of-school science settings are a productive site for science teachers to have opportunities to learn about their students and make relevant connections to their science learning.

Research on teacher learning in out-of-school settings highlights how these settings afford teachers opportunities for learning that are not regularly available to them in formal school settings. For example, pre-service science teachers are afforded opportunities to try out inquiry-based, hands-on learning experiences with youth in afterschool programs without the external pressures they face in their schooling contexts [3]. These low-stakes settings support pre-service teachers to experience teaching in new ways and the space to reflect on and consider how to incorporate this learning in their future classrooms. Further,

when science teachers see the value and make connections to students' learning experiences outside of formal classrooms, students and teachers benefit from expansive views of science learning beyond content to include interest and identity development in science [4].

In this study, we explore the learning opportunities for in-service science teachers when they participated in an out-of-school professional development (PD) program to design and implement a STEM after-school program (ASP) in partnership with a local university. Specifically, we look to understand the following:

1.  How did participation in the PD and implementation of an after-school program support science teachers to make science content more relevant to their students?
2.  What activities or tools from the PD and after-school program did the science teachers highlight as important for their learning?

## 2. Conceptual Framework

We take a situative perspective on science teacher learning [5,6]. Science teacher learning is embedded in complex and changing organizational environments, where they often have to make sense of and manage competing messages about science teaching and learning [7,8]. Therefore, when considering science teacher learning within the context of out-of-school learning environments, it becomes important to understand and articulate the possible sources of ambiguity and/or connections between in- and out-of-school science learning [9]. For example, school science education is organized around a set of common standards and assessments at the state level, whereas out-of-school science is oriented toward student choice and interest [10]. While these two objectives can appear to be in contradiction to one another, we argue that we can leverage science teachers' experiences in out-of-school settings to explicitly support teachers to make connections between students' interests and the science content they are teaching.

Most studies of science teachers in informal learning settings are focused on pre-service teaching and provide alternative spaces for pre-service teachers to observe and engage students in science learning that they might not see in formal classroom settings. For example, pre-service teachers might have internship experiences teaching inquiry in museum settings [11], develop and implement field trips in connection to the curriculum [12], plan and implement club activities at a school [3], or organize and facilitate a maker faire for their community [13] These informal settings provide pre-service teachers with opportunities to plan and try out different pedagogical practices and make connections across different learning settings [3]. For example, the study in [13] found that after planning and facilitating a maker activity, pre-service teachers highlighted both content understanding of youth as they engaged in the activity, but also their social and emotional learning evidenced through the activity.

Research on pre-service science teacher learning in these settings focuses on teacher development of content and pedagogical knowledge, their confidence in teaching science, and their perspectives on science teaching [14]. Further, other studies outside of science, have demonstrated how the role of observation within informal learning environments provides pre-service teachers with opportunities to expand their ideas about what learning looks like outside of the school walls [15]. In this study, we build on this literature base to focus on in-service science teacher learning to make science content relevant for students in out-of-school settings.

*STEM Learning as Relevant to Students*

There are multiple ways that relevance has been conceptualized in science education, including cultural relevance [16], socio-scientific issues [17], and personal relevance [18]. Across the literature base, there is wide variety in the ways in which relevance has been conceptualized [19]. In this paper, we define relevance as the pedagogical practice of leveraging local science content knowledge as a means to develop personal connections between teachers and students [20].

One program, STARTS (Science Teachers are Responsive to Students), involved supporting six high school science teachers to develop culturally responsive, reform-based teaching practices over the course of six months ([20], p. 104). The professional development focused on teachers' daily practices in parallel with science content to develop more culturally responsive pedagogical practices. As teachers implemented these culturally responsive pedagogical practices, four themes emerged as key to changing their pedagogy. These themes included a changed view of students, repositioning of power, community building within the classroom, and utilizing a culturally responsive pedagogy toolbox.

Initially, teachers' views of students were based on deficit stereotypes but as teachers developed relationships with students, their views of students changed. Repositioning of power was represented by teachers changing their pedagogical practices to include student-centered input. This looked like creating opportunities for students to share their knowledge on how they were perceiving science content. Community building within the classroom was developed by having spaces where students could openly share their ideas about the science content to build academic morale. Lastly, teachers used what they learned about their students to develop and build a toolbox of culturally responsive lessons. By engaging in this process, teachers were able to gain a better understanding of ways that they incorporate more culturally responsive science lessons, while being careful to not further essentialize students.

The next study took a more constructivist lens to look at how culturally relevant instruction developed with two teachers [21]. Naturalistic inquiry from observations of teachers' daily classroom activities and planning were used to develop the two case studies. The study found that restructuring interactions (authority), scaffolding of student's prior knowledge (achievement), and cultivating relationships and cultural interdependence (affiliation) were key to developing culturally relevant practices within science classrooms [21].

The teachers began to restructure interactions by supporting students' exploration with science content. However, there was no authentic redistribution of authority within the classroom. While the practices were implemented, they were not meaningfully demonstrated. Teachers also used scaffolding to acknowledge students' prior knowledge. However, when teachers scaffolded the science content they did not collaborate with students. Teachers made decisions that they felt were in the best interest of the student's achievement. Lastly relationships were evident and developed between the students and teachers. However, there was a lack of evidence with teachers incorporating students' native languages.

From the literature, we see the positive impacts of having relevant science content integrated into the curriculum. Having relevant science content within the classroom opens up the opportunities for student engagement and contribution. We also see how out-of-school learning experiences can support pre-service teachers to identify students' assets and resources for learning. We build on this literature base to understand how out-of-school settings can support science teachers to shift their perceptions about students and open up possibilities in their teaching practices.

## 3. Methods

To understand if and how the PD and implementation of the ASP supported science teachers to make science content more relevant to students, we used case study methodology [22]. We used a case study to develop an in-depth understanding of how two teachers developed an ASP that utilized their local content knowledge about water issues within the community. This case study specifically addresses impactful activities, ranging from participation in the PD to the implementation of their own ASP. Next, we describe the PD design, the ASP design developed by the teachers to describe the context of the science teachers' learning. Then we provide participant descriptions, data sources, and analytic methods.

*3.1. Design of Programs*

3.1.1. PD Design

The two teachers at the focus of this manuscript participated in a PD program to support their implementation of a STEM ASP as part of the design of a connected science learning system [23]. The NSF-funded project had multiple and intersecting programming designed to create more connected STEM learning for middle school youth. Programming for youth included a four-week Saturday Academy on the university campus followed by a six-week ASP designed and implemented by their teachers.

There were multiple objectives and learning goals for the teachers' participation in the project. Here, we focus on the project goal of supporting teachers to understand how to make STEM locally relevant to students. To support that goal, teachers participated in 45 h of PD over the course of two academic years, in parallel to their students' participation in out-of-school programming at the university. In this way, teachers could be introduced to the connected learning system first as observers and then as co-designers.

A central activity for the PD was supporting science teachers to understand the pedagogical framework that organized the connected science learning system. The pedagogical framework focused students on a local socioenvironmental problem and then supported them to use science, engineering, and technology to investigate and develop solutions for that problem [24]. The project team chose the issue of stormwater management and supported students to both understand the problem and its impacts on humans and the environment. Then they were introduced to green technology and design to minimize stormwater's negative impacts on humans and the environment and asked to design solutions for local stormwater issues.

Once the teachers were introduced to the pedagogical framework and how we developed activities based in that framework, teachers were invited to observe students as they engaged in activities led by university faculty and staff. We asked the teachers to listen to students as they engaged in the activities and then come back together and reflect on what they noticed and wondered about student engagement and learning in the activity. As teachers observed and interacted with students in these spaces, they started to notice and pay attention to resources and assets students brought to their learning [8]. After teachers experienced observing students and understanding how program activities aligned with the pedagogical framework, they began to design their ASP to continue their students' learning about stormwater management at their respective schools.

3.1.2. Design of the ASP

The ASP at Vista Middle School (VMS; names changed to pseudonyms) happened over the course of six weeks on Wednesdays directly after school for one hour in a designated classroom. The teachers designed the ASP to support the students to identify and understand stormwater management issues on their school campus and why they existed. While a key component of the pedagogical framework was to support students to develop solutions to these issues, the teachers feared that they would not have the resources or time to effectively manage the problem. Table 1 briefly describes each of the activities designed and implemented by the teachers in the ASP.

**Table 1.** Descriptions of the activities in the ASP.

| Week | Description of Activity | Objectives |
|---|---|---|
| 1 | Painted enviroscapes—painted a molded landscape that included the layout of a water source, farm, roadway, and mounds. | Used as an introduction to the club and connection with the university program |
| 2 | STEM profiles—students self-select into different STEM profile categories displayed across the room on posterboard, and discuss why they identified with that profile | Used to help students identify potential strengths in STEM, as well as build on their STEM profile surveys (taken previously) |
| 3 | Storm water sleuthing—students walk around their school after a rainy day to identify locations where stormwater collects | Used to locate issues with stormwater management at the school |

**Table 1.** *Cont.*

| Week | Description of Activity | Objectives |
|---|---|---|
| 4 | Card sort and Enviroscapes—students use their enviroscapes to create different stormwater management scenarios that are determined by the card they drew | Used to understand how different natural and man-made scenarios effect/affect stormwater management |
| 5 | Permeability—student used varying materials to determine the infiltration abilities and usability of porous/non-porous pavement | Used to understand infiltration abilities of different materials |
| 6 | Visit the creek—students used a water quality testing kit to determine pH, turbidity, dissolved oxygen, and temperature of a nearby creek | Used to determine the "health" of the water in the creek by the school |

*3.2. Participants*

This paper focuses on two teachers from one school, VMS, that participated in the PD and then co-designed and implemented the ASP at their school. VMS is an urban middle school where the student population is approximately 97% students of color. The ASP had an average of seven student participants which included four females and three males. The demographics of the students were Black (2), Hispanic (3), and white (2). Two science teachers led the club. Mr. Antonio, a white male, taught 7th-grade science and had been teaching at VMS for four years at the time of the study. He had a bachelor's of science in middle grades education with a focus on science and social studies. Prior to teaching, he was in construction and volunteered at his children's school in the gardens and greenhouse. He described his teaching as based on the 5E (engagement, exploration, explanation, elaboration, evaluation) model [25] and heavily focused on students' learning of science vocabulary. The second teacher, Mrs. Tanisha, a Black female, taught 8th-grade science and had been teaching at VMS for 10 years at the time of the study. She had a bachelor's and master's degree in biology and had taught in STEM summer camps at a local university as a student. She similarly followed the 5E [25] model for teaching science and said that she focused heavily on science content covered in the 8th grade state assessment. The first author, a Black female, facilitated activities with students during the Saturday Academies and was a participant observer during the ASP. The second author, a white female, designed and led the PD.

*3.3. Data Collected*

We collected video and audio recordings of the PD meetings with teachers and ASP meetings at VMS. We also audio recorded a debriefing conversation with teachers after each ASP meeting. In these conversations, we asked teachers what they thought went well, what challenges surfaced that day, and what resources or support they needed from the research team for the next club meeting. The data collected is reflective of 45 h of PD sessions from Spring 2018 to Fall 2019 and 8 ASP sessions where the teachers would debrief afterwards. We interviewed teachers four times over the three years of the project. The initial interview happened prior to the start of the PD and asked the teachers about their backgrounds, beliefs about science teaching and learning, and their experience teaching science both in- and out-of-school science learning. The subsequent interviews happened each summer to support teacher reflection of their learning, as well as to provide feedback to the research team about student programming and any support they needed from the research team.

*3.4. Data Analysis*

3.4.1. PD Data Analysis

We sampled conversations from the PD meetings that asked teachers to debrief and reflect on their observations of students' activities during the Saturday Academies. We used inductive coding and a constant comparative method [26] to analyze these conversations. Each author went through the transcripts and coded the focus teachers' reflections using the coding program Dedoose. For each utterance of a focus teacher, we noted the content

of their contribution, what they bracketed from student engagement in activities, and if and how they were connecting students' engagement with the science content. We wrote short research memos about the nature of the teacher post-observation conversations, as well as the tools from the PD that they leveraged as they made sense of student activity. We met and read through each other's research memos and came to a consensus about claims related to teacher learning [26]. We then reviewed interview data and researcher notes to confirm or disconfirm our claims about their experiences in the PD.

### 3.4.2. ASP Data Analysis

For this analysis, we used data collected during the ASP and interview data from teachers. As a participant observer of the ASP, the first author noticed that both the teachers and students talked about the importance of the time spent becoming acquainted with one another during the ASP. We wondered if these informal interactions in the ASP opportunities for the teachers were to make science content more relevant to their students. To identify these opportunities, we used interaction analysis [27] of the video data from the ASP. We made content logs of each video of the ASP meetings. These content logs captured the flow of activity, the nature of interactions between participants, and the material resources available. Each author reviewed the content logs to look for moments of informal dialogue between teachers, researchers, and students. As we watched, we each would stop the video and discuss moments where we noticed aspects of teachers making connections between science content and students' lives. We wrote research memos summarizing our claims about how these informal interactions reflected moments of science teachers making connections between what they were learning about their students and the science content to adjudicate disagreements. Finally, we used audio data from debriefing meetings and interviews with the teachers to look for confirming or disconfirming evidence.

## 4. Findings

We found that the PD supported the science teachers to develop three mechanisms to make connections between students' lived experiences and their STEM learning. First, the teachers were able to build different kinds of relationships with students than they were able to in the classroom. Second, the teachers developed what they learned about students' lives to make connections to the science content. Third, these relationships supported teachers to disrupt school-based deficit narratives around youth of color. In the following sections, we answer each research question to demonstrate the PD tools and ASP activities that supported science teacher learning.

### *4.1. Science Teacher Learning in PD*

Our first research question focused on understanding how science teachers' participation in PD in an out-of-school context supported them to make the science content more relevant to their students in their ASPs. Over the course of the PD the teachers began to notice how the students were making connections between the content they were learning and the world around them. Three features of PD supported this learning. First, the observations of students' interactions in the activities provided the teachers to see and listen for students' sensemaking as they engaged in the activities. Second, the debriefing conversations after the observations supported the teachers to make connections between what they observed during the out-of-school science activities to what they did in the classroom. Third, the students' STEM identity profile and the pedagogical framework provided by the research staff were important tools that mediated science teacher sensemaking of what they observed.

In the first year of their PD, the two focus teachers made connections between student engagement in the out-of-school setting and their behaviors at school. Interestingly, they surfaced and discussed behaviors that were celebrated in the out-of-school setting and were seen as deficits in the classroom. Here is an example from their first PD, the teachers are debriefing observations of students down at the wetland where they were exploring

the wetland with a variety of different science instruments. They were talking about one student who jumped right into the activity and did not care about becoming dirty.

**Mr. Antonio:** So, the first young man that you guys were talking about, that jumped in, is he one too that is very messy? Is he organized? Or is he someone that is. So, this is what I'm picturing. He's reminding me of one of mine [teachers talking over one another].

**Mrs. Mary:** Probably didn't take any actual notes. You'd be lucky if he did have his pencil or a bookbag.

**Mr. Antonio:** Okay. Because I have a few like him...If he actually has the notebook . . . But I mean he is one too that every time he takes his test it's a 95 or 100. So, he's getting it. So, you're going to have to be like okay I got to let some of this stuff go that I really need him to do, but he's getting it.

Here, Antonio was making a connection between a positive behavior in the out-of-school activity, getting dirty, and being disorganized at school. He went further to note that even though the student is disorganized, they were successful at learning science. During the same conversation, Tanisha was making connections between the students' STEM identity profile and their engagement at the wetlands. She began by wondering what STEM profile a group of female students to connect to their behavior.

**Mrs., Tanisha:** I was wondering about the student profiles, when you saw them doing specific things or you know, their interest. How they work together. There was one group I saw working together, I was wondering what their profiles were like . . . But seeing how they interact together, I just continue to wonder "what's their profiles like?

Here, Tanisha was trying to understand student engagement at the wetland through their identities as science doers. The PD provided a more expanded view of what it meant to do science and Tanisha leveraged this tool to make sense of the students' activities. During the second year of the PD, the teachers began making more connections between the pedagogical framework and student engagement. Antonio focused on the need for the activities to not look like school and even critiqued some of the revisions the project staff had made to the activities, noting that it looked more like school.

**Mr. Antonio:** Well, how can you hook them that first day? How can you make sure that they know this isn't school?

**Mrs. Vanessa:** And I think it will be interesting for us to listen on the way home to the conversations because they said several times in our groups, 'this isn't school, you know. It is not school', but it still felt like school to them where I think down at the creek that doesn't necessarily mean-

**Mr. Antonio:** [interrupts Vanessa] Exactly. I was just getting ready to say, I think the kids that were inside kind of felt like, 'well wait a minute. He told me this wasn't going to be like school'.

This idea in the pedagogical framework that students needed to go outside and do science became important for Antonio as he planned the ASP at his school site. He noted that students were able to make strong connections between what they were learning and what they observed outside and wanted to maintain that connection. In the next section, we describe how this learning translated into the ASP setting and supported the teachers to make the content relevant to the students through an attention to their STEM identity profiles and outdoor activities where students could connect what they learned to what they observed.

*4.2. STEM Profiles*

At the beginning of the ASP, students were given a survey to discover their possible STEM identity. The STEM Identity profile instrument is an inventory-style survey that

yields a combination of six STEM-related affiliations that include tinkerer, conservationist, altruist, investigator, inventor, and designer [9]. Tinkerers work well with their hands and enjoy building and making tangible products or solutions. Conservationists care about the impact of humans in the environment and advocate for sustainable change. Altruists care deeply about people, their health, and their safety. Investigators enjoy asking questions and synthesizing information to find patterns or themes. Inventors enjoy working through complex puzzles, logic games, and strategizing solutions for problems. Designers are very visual and have a keen eye for beauty and aesthetics and enjoy creating works of art in many forms.

The STEM profile activity was an extension of the results of students' STEM identity profiles. The teachers placed six large poster sheets around the room to represent the STEM-related affiliations (tinkerer, conservationist, altruist, investigator, inventor, and designer). Students were given six labels with their names on them to place on posters they felt they identified with the most. For example, if a student felt strongly about being a tinkerer, they could place three or all six of the labels on that poster. This served as a way for students to self-identify with the different STEM-related affiliations and to make sense of the various ways that people can identify with STEM.

Mr. Antonio was partnered with two other students, Carmen and Manuel, and they discussed why they felt they were tinkerers. Below is an excerpt from their conversation.

**Mr. Antonio:** Why did you guys choose this one (tinkerer)?

**Manuel:** Because we like building.

**Mr. Antonio:** What [what do you like building]?

**Carmen:** Houses, anything.

**Mr. Antonio:** Houses [to confirm]? Okay, this is what I used to do. Primarily this is me. I work with my hands. I like to build, make, vent, fix, take apart.

**Manuel:** Me too.

**Mr. Antonio:** So, is this primarily where you two, like do you do this or is this just where you put yourself [identifying with tinkerer]?

**Manuel:** I used to do it, over the summer.

**Mr. Antonio:** So, you've done what, construction [to clarify]?

**Manuel:** Yeah.

**Carmen:** Yeah, same.

**Mr. Antonio:** And you enjoyed building stuff with your hands. Was it the same with you [directed towards Carmen]?

**Carmen:** Yeah, during breaks and summer.

**Mr. Antonio:** During what now [asking for clarification]?

**Carmen:** Like during spring break and summer.

**Mr. Antonio:** So, construction, same thing?

**Carmen:** Yes

**Mr. Antonio:** Is that because of parents?

**Manuel:** No because we love it.

**Mr. Antonio:** Oh okay, have you ever just taken stuff apart to see how it works? [Carmen and Manuel nod heads to answer yes] Can you put it back together and it still works?

**Carmen and Manuel:** Sometimes

**Mr. Antonio:** Yeah, I used to have that problem. I used to take things apart to figure out if I could fix it. Sometimes I could, sometimes I couldn't. I like definitely to see the tangible product. I like to see it, work with it, manipulate it, and that's why I'm not a big designer, myself.

As Carmen and Manuel explained why they identified as being tinkerers, Mr. Antonio learned about their background experience with construction. Once Mr. Antonio became aware of Carmen and Manuel's construction experience, in conversations with others, he would always highlight their construction background and position them as experts. Mr. Antonio created a space for Carmen and Manuel to connect their lived experiences with construction to science content.

From the examples above, we see how the ASP provided opportunities for students to make connections between their interests and lived experiences with science content. The ASP provided an environment where teachers could acquaint themselves with their students by building personal relationships. Teachers usually only become familiar with their students as "the person in my class", but during the ASP, teachers learned about student backgrounds and interests. Students came with a wealth of background knowledge that they were not always able to express in their classrooms. Mr. Antonio and Mrs. Tanisha created space for students to be open, allowing them to recognize the students' authentic selves. Through their dialogue, the teachers were able to authentically learn about their lives.

*4.3. Stormwater Sleuthing*

Stormwater sleuthing is an interactive walk where participants identify areas where stormwater collects, assess management practices of water, and collect visual data of stormwater areas. This activity was adapted from materials by [28] and organized around three questions; (1) Where does the water go? (2) Where are potential stormwater management opportunities? and (3) How can we solve those problems? The ASP took place from February to April of 2019 and there was a lot of rainfall. This made the environmental conditions ideal to engage in this activity. The stormwater sleuthing activity at VMS demonstrated how conversations during this activity supported Mr. Antonio and Mrs. Tanisha to make connections between students' lived experiences and the science content.

Mr. Antonio took the lead during stormwater sleuthing and guided students around the school. Due to his construction background, he was able to create in-depth connections to the science curriculum and localized, real-world examples. For example, youth had identified the walkway from the gymnasium to the cafeteria as a major problem spot for stormwater. Students collectively expressed that when it would rain this walkway would be flooded causing many people's feet to become drenched. During the stormwater sleuthing activity when Mr. Antonio approached this area he posed the question to students, what are you noticing about this area (that contributes to flooding). Manuel pointed to one of the column supports that connected to the roof of the gymnasium. This column had a hole in it near the bottom for water to drain from the roof and enter into the storm drain in the ground. Mr. Antonio asked students about their thoughts on why the water was not reaching the drain. Most students identified that the hole in the column was clogged with natural debris, e.g., leaves and twigs. Mr. Antonio agreed that the clogged column was a part of the problem, but also pointed out that the ground did not have the proper incline for water to reach the storm drain. He shared with students that the janitorial staff at the school tried to fix it but did not do it correctly. The janitorial staff had used a shovel to make a pathway for the water to flow into the storm drain. However, this still did not solve the problem as the pathway still did not have the proper incline for stormwater to follow.

As Mr. Antonio led the stormwater sleuthing activity, Carmen walked with Mrs. Tanisha and pointed out what she noticed about how the landscape of the school grounds were affecting where the stormwater flowed. Carmen used her hands to model the landscape of what she noticed. As Carmen modeled, she stood beside Mrs. Tanisha and leaned in towards her to explain how the landscape design was leading to flooding. Mrs. Tanisha

asked Carmen about her solution, and she began to once again model with her hands and show that the landscape should change so water could flow through it easier. Mr. Antonio, in the background, was explaining why one area of the school continually floods due to a clogged drain. Mrs. Tanisha asked Carmen to repeat her idea for the others. The group did not initially hear Carmen's explanation, leading Mrs. Tanisha to speak up louder and ask Carmen to say what she just said again. This time it engaged another student, Manuel, to listen and share his idea also. Manuel and Carmen were now discussing how they could improve the stormwater issues. As the group continued with sleuthing Carmen would come back to Mrs. Tanisha to point out other landscape issues in different locations. Each time Mrs. Tanisha would pause the group by asking Carmen to say what she just said. This forced the group to give Carmen attention as she explained how the landscape affected the stormwater runoff.

As we can see through these examples, the teachers were able to provide opportunities to build connections between students' lived experiences and how that contributed to their science content knowledge. Within their classrooms, students may not receive the opportunity to share their lived experiences. In the ASP, Mr. Antonio made sure that students were able to speak up and share what they knew while catering to their localized content knowledge and leveraging students' funds of knowledge [29]. Mrs. Tanisha created a "shared intellectual space" with Carmen through their dialogue as they walked around the school's campus. Carmen shared her knowledge about landscape with Mrs. Tanisha and then Mrs. Tanisha positioned her as an expert by inviting her to share this information with the group. Mrs. Tanisha did this twice to ensure that Carmen was the person sharing her ideas. The first time Mrs. Tanisha did this, Manuel was invited into the shared intellectual space where he also contributed his knowledge.

*4.4. Reflections/Debriefs*

About midway through the ASP, Author 1 worked with Mr. Antonio and Mrs. Tanisha to finalize their plans for the remainder of the ASP. During this meeting, there was a short survey to ask how the ASP had been going thus far. Author 1 asked for a memorable moment from the ASP so far. Mrs. Tanisha had expressed how she was unable to engage with her students on a more personal level within the classroom because of the heavy science context environment. Below is Mrs. Tanisha's memorable moment.

> "I don't have great rapport with my students the way I want to because I'm so (science) content, content, content. Like Manuel comes up to me all the time now (during school) and we don't even have that type of rapport and he comes up to talk about things from after school (ASP). So, it helps me to know him on a more personal level."

When we finished the debrief about how the ASP was going so far, we then shifted into planning. Both teachers wanted to have the next five sessions planned out before they became consumed with EOG (state science test) preparation. As we were discussing planning, Mr. Antonio shared an experience that he had with students (both participants and non-participants of the ASP) during the school day. It had rained recently and because students had recently engaged in the stormwater sleuthing activity, they wanted to share their new findings on the school grounds. Below is a transcript of the conversation between both teachers and Author 1.

> **Mr. Antonio:** Oh yesterday, you don't know how many of them, seventh and eighth graders were wanting to go show me stuff. Where new lake things [large puddles of water] had appeared. Where all the grates were now clogged, like they showed me. I never even realized that out behind my [classroom] window it gets so bad, the grate is clogged, and water fills up all behind that area. They [students] were like 'that whole grate is clogged again Mr. Antonio!'
>
> **Mrs. Tanisha:** (laughs) Aww, that's so sweet.

**Mr. Antonio:** Because I have that window open the eighth graders will come to it.

**Mrs. Tanisha:** Really?

**Mr. Antonio:** Uh huh.

**Mrs. Tanisha:** To talk to you?

**Mr. Antonio:** Uh huh.

**Mrs. Tanisha:** What do they say, do they just say hey Mr. Antonio?

**Mr. Antonio:** Oh yeah, like they yell in the window during class change.

**Mrs. Tanisha:** And then they keep going?

**Mr. Antonio:** But yesterday was about, 'there's a lot of water out here, this grates clogged'.

[Mr. Antonio and Mrs. Tanisha both laughed at the students' enthusiasm.]

**Author 1:** Was it students from the club (ASP)?

**Mr. Antonio:** Yeah, about two or three of them, and then some that weren't in the club that were just walking around with them.

**Author 1:** Oh okay. (laughs)

From the above conversations, we can see that the ASP offered teachers an opportunity to see their students with a different lens. Students were now seeking engagement with their teachers beyond the ASP about the science content they were learning in the club. By experiencing the ASP with youth, the teachers were able to disrupt and challenge the school-based deficit narrative around youth of color—primarily, that they are not interested in science or STEM. The ASP opened opportunities for teachers to see how their students navigated science in different settings outside of their own classroom.

## 5. Discussion

This paper explored how participation in PD and implementation of an ASP supported science teachers in making content more relevant for their students and which activities and tools were important for their learning. While a majority of research on science teacher PD focuses on changes to science teachers' classroom practices to support reform-oriented science learning [30], the findings show support for leveraging informal settings. The two teachers described school-based constraints that limited shifts to their classroom practices. Teachers instead saw the ASP as two things: first, a site where they could leverage more hands-on science activities based on relevant community-based socioenvironmental issues [31]; and second, a site to develop relationships and rapport with students [32,33]. We argue that the ASP afforded teachers opportunities to engage in equity-based teaching practices when they were constrained by school-based practices focused on passing state science assessments [34,35]. As mentioned above, equity-based pedagogies usually focus on leveraging students' assets and resources for science learning in formal settings; however, these findings provide insight on what this could look like in informal settings [36,37]. Mainly the ASP highlighted the need for opportunities to work with students in informal settings and the importance of developing student–teacher relationships.

### 5.1. Provide Opportunities to Work with Students in Informal Settings

The findings presented made an argument that informal science learning settings were named as productive and even transformative spaces for science teachers. This work builds off other research on pre-service teachers learning in informal settings [3,9,11] and demonstrates that the teachers were able to identify students' assets and resources for learning in the out-of-school setting [36]. For example, when Ms. Tanisha and Mr. Antonio talked with students during the STEM profile activity, it offered a lens for them to see how students saw themselves in STEM and why. As students made connections between their

daily lives and the stormwater content, both teachers were able to create opportunities for students to share and build on that knowledge within the ASP. Similarly, pre-service teachers working in an afterschool setting were able to recognize students' everyday literacy practices and make connections to the academic literacy practices discussed in their education courses [15].

In the debrief, Ms. Tanisha stated that because of the heavy focus on science content within her classroom, it did not provide many opportunities for her to really engage with students outside of science content. The informal setting was critical for Ms. Tanisha because now that she was not limited to focusing solely on content, she could take more of an inquiry-based approach. Science teachers frequently discuss the tension between teaching in inquiry-based ways and covering content to pass state tests [38,39]. Results from this study suggest that connections with students outside of the accountability context support science teachers to experience teaching and learning with their students in new ways. This is also highlighted in studies with pre-service teachers where they also identified the need to have opportunities to try inquiry-based science teaching in informal settings away from the system of accountability that constrained their formal classrooms [4,14].

### 5.2. Need for Relevant Science Activities

The focus on stormwater issues in the ASPs established a personal and relevant connection between both teachers and students [16–18]. The students were familiar with the effects of stormwater because they had experienced these effects within their communities (i.e., school and home) [31]. During the ASP, Mr. Antonio and Mrs. Tanisha were able to leverage students' familiarity with the impacts of storms on their everyday life in a way that provided students more agency in participation in disciplinary practices, built on students' funds of knowledge, and created new forms of cultural interdependence [21,30,40]. Mr. Antonio and Mrs. Tanisha increased students' agency to develop solutions to local stormwater issues. These interactions led to cultural interdependence as Mr. Antonio, Mrs. Tanisha and the student participants relied on each other's expertise to create new solutions for their school. Through this experience, the teachers learned to consider student agency in their learning when designing learning opportunities [40]. The above example showed the importance of teachers observing and working alongside students within informal settings to shift how teachers view their students.

Both teachers were able to develop a culturally responsive pedagogy toolbox to support student learning of stormwater [20]. Students were expected to learn and leverage their understanding of key science content related to water and land interactions, as well as engineering and technology content and practices to develop solutions to locally meaningful stormwater issues. The focus on STEM content and practices supported students' academic achievement in the context of stormwater, a key component of the 8th-grade curriculum. They learned this content as Mr. Antonio and Mrs. Tanisha affirmed and respected students' cultural knowledge about stormwater. Lastly, as students learned about how decisions by the school and city led to stormwater issues within their community, they were able to broaden their critical consciousness about the complexities between stormwater and social justice [41].

### 5.3. Importance of Student–Teacher Relationships

Mr. Antonio and Mrs. Tanisha did not lose their focus on students' development of content knowledge in the ASP. They regularly made connections to science content assessed in the 8th-grade state assessment; however, it was not centered in their discussions. Rather, the students and their experiences were centered and shared, which shifted teachers' perceptions about students [20]. This is highlighted in the reflection/debrief had with Mr. Antonio and Mrs. Tanisha. This is similar to studies where teachers' perceptions of students change as they develop relationships with their students [18,20,31].

In the ASP, a community was developed by taking a more culturally relevant pedagogical approach. Mr. Antonio and Mrs. Tanisha created a space where students could

openly share ideas about science to increase their academic morale [20,21]. However, the ASP also served as a humanizing space that supported students to be their authentic selves and welcomed their lived experiences to support their learning [42]. By reducing power dynamics, students were able to engage in meaningful conversations with Mrs. Tanisha and Mr. Antonio to develop relationships.

This work seeks to address a void in in-service teacher professional development literature by specifically considering how relevant science content can support student learning. It has implications for in-service science educators who seek to explore how implementing relevant science content can also provide opportunities to develop meaningful relationships between students and teachers. By placing value on other outcomes besides science content knowledge, teachers can engage in equity-based teaching practices.

## 6. Limitations

In this section, we acknowledge limitations of this study. First, because we used case study methodology to focus on two teachers, it made the data harder to generalize. Second, while teachers were implementing a second session of the ASP, we were halted by the COVID-19 pandemic. This meant we were only able to collect data from one implementation of the ASP.

## 7. Conclusions

This paper provides insight into the various ways in which STEM out-of-school contexts can be beneficial for teachers' efforts to develop relevant science content. When teachers discussed stormwater, they provided examples that were directly connected to their school, neighborhoods, and city. However, by centering student's lives within the ASP, teachers were able to find value in other outcomes such as developing relationships with students. Looking forward, it is important to consider the opportunities available for in-service teachers learning in informal settings that are not afforded by formal classroom constraints. While the majority of the literature focuses on the experiences of pre-service teachers within informal settings, we contribute by looking at the experiences of in-service teachers—specifically, how informal settings can support science teachers' formal classroom practices. Informal settings can provide new opportunities for teachers to improve their practices and integrate that new material into their classrooms. However, it also provides opportunities for teachers to develop deeper relationships and rapport with their students.

**Author Contributions:** Conceptualization, T.W. and S.H.; methodology, T.W. and S.H.; software, S.H.; validation, T.W. and S.H.; formal analysis, T.W. and S.H.; investigation, T.W. and S.H.; resources, T.W. and S.H.; data curation, T.W. and S.H.; writing—original draft preparation, T.W. and S.H..; writing—review and editing, T.W. and S.H.; visualization, T.W. and S.H.; supervision, S.H..; project administration, S.H.; funding acquisition, S.H. All authors have read and agreed to the published version of the manuscript.

**Funding:** This research was funded by the National Science Foundation grant number 1657194.

**Institutional Review Board Statement:** The study was approved by the Institutional Review Board of the University of North Carolina at Greensboro (protocol code 16-0392, May 2017).

**Informed Consent Statement:** Informed consent was obtained from all subjects involved in the study.

**Data Availability Statement:** The data presented in this study are available on request from the corresponding author. The data are not publicly available at the participants' request.

**Conflicts of Interest:** The authors declare no conflict of interest.

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
