# Peer review of "Exploring STEM Teacher Learning in Out-of-School Settings: Sites for Learning about Relevance"

_education, doi:10.3390/educsci13030305_

Round 1

Reviewer 1 Report

I question the identification of the race and genders of the teachers involved. There was not any reference to how that impacted results, connections to background research, or the findings itself. Using other names for the teachers that do not identify gender or other cultural information would be best.

typo on line 130

error on link to table 1 in section 3.1.2; line 184

punctuation error on line 213

line 243 - suggestions for a different verb such as "established, built or developed"

Author Response

Feedback: I question the identification of the race and genders of the teachers involved. There was not any reference to how that impacted results, connections to background research, or the findings itself. Using other names for the teachers that do not identify gender or other cultural information would be best.

Response: The race and genders of the teachers were provided as a means to provide context about whom was involved. There was not intent to suggest that this impacted the results. Pseudonyms are provided for the teachers.

Feedback: typo on line 130

Response: The word having was removed.

Feedback: error on link to table 1 in section 3.1.2; line 184

Response: The link has been fixed and now works.

Feedback: punctuation error on line 213

Response: The period was removed.

Feedback: line 243 - suggestions for a different verb such as "established, built or developed"

Response: Verb changed from “were able to use” to developed

Reviewer 2 Report

This paper is very interesting, I appreciate the possibility to evaluate and comment on it. My comments are intended to improve the authors' contributions.

The comments are made following the order and structure of the manuscript.

The authors should structure the abstract including the sample by describing somewhat more the subject of the case study they present.

Although it is a well-known STEM term, do not use acronyms without indicating their meaning the first time they use it.

They should specify some more relevant data of the subjects who participated, for example; age, years of experience, and characteristics of the center where they teach..... I find it curious that you highlight the color of the skin and do not talk about variables such as professional experience, and motivation toward teaching. It would be interesting to have this information available and I recommend eliminating the allusions to skin color.

How did you validate the responses and observations made, was an inter-rater content analysis carried out and was any qualitative analysis program used?

The discussion they present is not a real discussion. They should discuss the results of the study in light of the theoretical development in the introduction of the manuscript.

The ethical aspects of this study are not addressed. Did any ethics committee approve this study? How was the confidentiality of the data guaranteed, how was the consent of the participants obtained, and did they all disagree to participate? The ethical aspects of the study are not addressed.

The authors should include a section on the limitations of the study.

The conclusions section should be improved, not only focusing on the conclusions in terms of the experience but the implications for the improvement of the teaching practice of science teachers.

Author Response

Feedback: This paper is very interesting, I appreciate the possibility to evaluate and comment on it. My comments are intended to improve the authors' contributions.

Response: Thank you.

Feedback: The authors should structure the abstract including the sample by describing somewhat more the subject of the case study they present.

Response: The abstract has been updated to include more information.

Feedback: Although it is a well-known STEM term, do not use acronyms without indicating their meaning the first time they use it.

Response: Description for STEM has been added.

Feedback: They should specify some more relevant data of the subjects who participated, for example; age, years of experience, and characteristics of the center where they teach..... I find it curious that you highlight the color of the skin and do not talk about variables such as professional experience, and motivation toward teaching. It would be interesting to have this information available and I recommend eliminating the allusions to skin color.

Response: The participants section has been expanded to include more information about the teachers background as well as information about the school and student participants. 

Feedback: How did you validate the responses and observations made, was an inter-rater content analysis carried out and was any qualitative analysis program used?

Response: The data analysis section was updated to provide clarifying information 

Feedback: The discussion they present is not a real discussion. They should discuss the results of the study in light of the theoretical development in the introduction of the manuscript.

Response: A subsection was added to the discussion to strengthen the need for more relevant science activities. Other subsections were edited to include stronger connections between the conceptual framework and the discussion.

Feedback: The ethical aspects of this study are not addressed. Did any ethics committee approve this study? How was the confidentiality of the data guaranteed, how was the consent of the participants obtained, and did they all disagree to participate? The ethical aspects of the study are not addressed.

Response: The information surrounding the ethics of this study were included during the submission process for this journal. This information will be included in the article pending acceptance.

Feedback: The authors should include a section on the limitations of the study.

Response: A limitations section was added.

Feedback: The conclusions section should be improved, not only focusing on the conclusions in terms of the experience but the implications for the improvement of the teaching practice of science teachers.

Response: The conclusion has been updated to better explain how this paper adds to the literature.

Round 2

Reviewer 2 Report

Thanks to the authors for addressing the suggested changes. The manuscript is now clearer and of higher quality.

In the abstract, clarify in parentheses the term STEM. The correct form to do is to put in meaning and in parentheses the acronym. Correct this and remove from the introduction the meaning of STEM. It repeats it in line 24. Please correct this

Delete reference 9 in the conclusion section. Authors should not be cited in the conclusions, this is done in the discussion.

Remove this citation and connect the text appropriately.

Author Response

Feedback: Thanks to the authors for addressing the suggested changes. The manuscript is now clearer and of higher quality.

Response: Thank you for your feedback and suggestions.

Feedback: In the abstract, clarify in parentheses the term STEM. The correct form to do is to put in meaning and in parentheses the acronym. Correct this and remove from the introduction the meaning of STEM. It repeats it in line 24. Please correct this

Response: The correct version has been applied in the abstract and explanation has been removed from line 24.

Feedback: Delete reference 9 in the conclusion section. Authors should not be cited in the conclusions, this is done in the discussion.

Response: The citation has been removed.

Feedback: Remove this citation and connect the text appropriately.

Response: Sentences have been restructured to connect text properly.